# Learning Large-Scale Poisson DAG Models based on OverDispersion Scoring

**Gunwoong Park**
Department of Statistics
University of Wisconsin-Madison
Madison, WI 53706
parkg@stat.wisc.edu

**Garvesh Raskutti**
Department of Statistics
Department of Computer Science
Wisconsin Institute for Discovery, Optimization Group
University of Wisconsin-Madison
Madison, WI 53706
raskutti@cs.wisc.edu

## Abstract

In this paper, we address the question of identifiability and learning algorithms for large-scale Poisson Directed Acyclic Graphical (DAG) models. We define general Poisson DAG models as models where each node is a Poisson random variable with rate parameter depending on the values of the *parents* in the underlying DAG. First, we prove that Poisson DAG models are identifiable from observational data, and present a polynomial-time algorithm that learns the Poisson DAG model under suitable regularity conditions. The main idea behind our algorithm is based on *overdispersion*, in that variables that are conditionally Poisson are overdispersed relative to variables that are marginally Poisson. Our algorithms exploits overdispersion along with methods for learning sparse Poisson undirected graphical models for faster computation. We provide both theoretical guarantees and simulation results for both small and large-scale DAGs.

## 1 Introduction

Modeling large-scale multivariate count data is an important challenge that arises in numerous applications such as neuroscience, systems biology and amny others. One approach that has received significant attention is the graphical modeling framework since graphical models include a broad class of dependence models for different data types. Broadly speaking, there are two sets of graphical models: (1) undirected graphical models or Markov random fields and (2) directed acyclic graphical (DAG) models or Bayesian networks.

Between undirected graphical models and DAGs, undirected graphical models have generally received more attention in the large-scale data setting since both learning and inference algorithms scale to larger datasets. In particular, for multivariate count data Yang et al. [1] introduce undirected Poisson graphical models. Yang et al. [1] define undirected Poisson graphical models so that each node is a Poisson random variable with rate parameter depending only on its neighboring nodes in the graph. As pointed out in Yang et al. [1] one of the major challenges with Poisson undirected graphical models is ensuring global normalizability.

Directed acyclic graphs (DAGs) or Bayesian networks are a different class of generative models that model directional or causal relationships (see e.g. [2, 3] for details). Such directional relationships naturally arise in most applications but are difficult to model based on observational data. One of the benefits of DAG models is that they have a straightforward factorization into conditional distributions [4], and hence no issues of normalizability arise as they do for undirected graphical models as mentioned earlier. However a number of challenges arise that make learning DAG models often impossible for large datasets even when variables have a natural causal or directional structure.

These issues are: (1) identifiability since inferring causal directions from data is often not possible; (2) computational complexity since it is often computationally infeasible to search over the space of DAGs [5]; (3) sample size guarantee since fundamental identifiability assumptions such as faithfulness are often required extremely large sample sizes to be satisfied even when the number of nodes $p$ is small (see e.g. [6]).

In this paper, we define Poisson DAG models and address these 3 issues. In Section 3 we prove that Poisson DAG models are identifiable and in Section 4 we introduce a polynomial-time DAG learning algorithm for Poisson DAGs which we call OverDispersion Scoring (ODS). The main idea behind proving identifiability is based on the *overdispersion* of variables that are conditionally Poisson but not marginally Poisson. Using overdispersion, we prove that it is possible to learn the causal ordering of Poisson DAGs using a polynomial-time algorithm and once the ordering is known, the problem of learning DAGs reduces to a simple set of neighborhood regression problems. While overdispersion with conditionally Poisson random variables is a well-known phenomena that is exploited in many applications (see e.g. [7, 8]), using overdispersion has never been exploited in DAG model learning to our knowledge.

Statistical guarantees for learning the causal ordering are provided in Section 4.2 and we provide numerical experiments on both small DAGs and large-scale DAGs with node-size up to 5000 nodes. Our theoretical guarantees prove that even in the setting where the number of nodes $p$ is larger than the sample size $n$, it is possible to learn the causal ordering under the assumption that the degree of the so-called moralized graph of the DAG has small degree. Our numerical experiments support our theoretical results and show that our ODS algorithm performs well compared to other state-of-the-art DAG learning methods. Our numerical experiments confirm that our ODS algorithm is one of the few DAG-learning algorithms that performs well in terms of statistical and computational complexity in the high-dimensional $p > n$ setting.

## 2   Poisson DAG Models

In this section, we define general Poisson DAG models. A DAG $G = (V, E)$ consists of a set of vertices $V$ and a set of directed edges $E$ with no directed cycle. We usually set $V = \{1, 2, \ldots, p\}$ and associate a random vector $(X_1, X_2, \ldots, X_p)$ with probability distribution $\mathbb{P}$ over the vertices in $G$. A directed edge from vertex $j$ to $k$ is denoted by $(j, k)$ or $j \to k$. The set $\text{Pa}(k)$ of *parents* of a vertex $k$ consists of all nodes $j$ such that $(j, k) \in E$. One of the convenient properties of DAG models is that the joint distribution $f(X_1, X_2, ..., X_p)$ factorizes in terms of the conditional distributions as follows [4]:

$$f(X_1, X_2, ..., X_p) = \Pi_{j=1}^{p} f_j(X_j | X_{\text{Pa}(j)}),$$

where $f_j(X_j | X_{\text{Pa}(j)})$ refers to the conditional distribution of node $X_j$ in terms of its parents. The basic property of Poisson DAG models is that each conditional distribution $f_j(x_j | x_{\text{Pa}(j)})$ has a Poisson distribution. More precisely, for Poisson DAG models:

$$X_j | X_{\{1,2,...,p\} \setminus \{j\}} \sim \text{Poisson}(g_j(X_{\text{Pa}(j)})), \tag{1}$$

where $g_j(.)$ is an arbitrary function of $X_{\text{Pa}(j)}$. To take a concrete example, $g_j(.)$ can represent the link function for the univariate Poisson generalized linear model (GLM) or $g_j(X_{\text{Pa}(j)}) = \exp(\theta_j + \sum_{k \in \text{Pa}(j)} \theta_{jk} X_k)$ where $(\theta_{jk})_{k \in \text{Pa}(j)}$ represent the linear weights.

Using the factorization (1), the overall joint distribution is:

$$f(X_1, X_2, ..., X_p) = \exp\Big( \sum_{j \in V} \theta_j X_j + \sum_{(k,j) \in E} \theta_{jk} X_k X_j - \sum_{j \in V} \log X_j! - \sum_{j \in V} e^{\theta_j + \sum_{k \in \text{Pa}(j)} \theta_{jk} X_k} \Big).$$

$$(2)$$

To contrast this formulation with the Poisson undirected graphical model in Yang et al. [1], the joint distribution for undirected graphical models has the form:

$$f(X_1, X_2, ..., X_p) = \exp\Big( \sum_{j \in V} \theta_j X_j + \sum_{(k,j) \in E} \theta_{jk} X_k X_j - \sum_{j \in V} \log X_j! - A(\theta) \Big), \tag{3}$$

where $A(\theta)$ is the log-partition function or the log of the normalization constant. While the two forms (2) and (3) look quite similar, the key difference is the normalization constant of $A(\theta)$ in (3) as opposed to the term $\sum_{j \in V} e^{\theta_j + \sum_{k \in \mathbf{Pa}_{(j)}} \theta_{kj} X_k}$ in (2) which depends on $X$. To ensure the undirected graphical model representation in (3) is a valid distribution, $A(\theta)$ must be finite which guarantees the distribution is normalizable and Yang et al. [1] prove that $A(\theta)$ is normalizable if and only if all $\theta$ values are less than or equal to $0$.

## 3 Identifiability

In this section, we prove that Poisson DAG models are identifiable under a very mild condition. In general, DAG models can only be defined up to their Markov equivalence class (see e.g. [3]). However in some cases, it is possible to identify the DAG by exploiting specific properties of the distribution. For example, Peters and Bühlmann prove that for Gaussian DAGs based on structural equation models with known or the same variance, the models are identifiable [9], Shimizu et al. [10] prove identifiability for linear non-Gaussian structural equation models, and Peters et al. [11] prove identifiability of non-parametric structural equation models with additive independent noise. Here we show that Poisson DAG models are also identifiable using the idea of overdispersion.

To provide intuition, we begin by showing the identifiability of a two-node Poisson DAG model. The basic idea is that the relationship between nodes $X_1$ and $X_2$ generates the overdispersed child variable. To be precise, consider all three models: $\mathcal{M}_1 : X_1 \sim \text{Poisson}(\lambda_1)$, $X_2 \sim \text{Poisson}(\lambda_2)$, where $X_1$ and $X_2$ are independent; $\mathcal{M}_2 : X_1 \sim \text{Poisson}(\lambda_1)$ and $X_2|X_1 \sim \text{Poisson}(g_2(X_1))$; and $\mathcal{M}_3 : X_2 \sim \text{Poisson}(\lambda_2)$ and $X_1|X_2 \sim \text{Poisson}(g_1(X_2))$. Our goal is to determine whether the underlying DAG model is $\mathcal{M}_1, \mathcal{M}_2$ or $\mathcal{M}_3$.

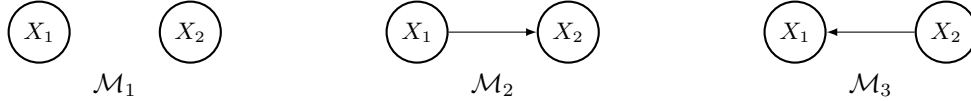

Figure 1: Directed graphs of $\mathcal{M}_1$, $\mathcal{M}_2$ and $\mathcal{M}_3$

Now we exploit the fact that for a Poisson random variable $X$, $\text{Var}(X) = \mathbb{E}(X)$, while for a distribution which is a conditionally Poisson, the variance is overdispersed relative to the mean. Hence for $\mathcal{M}_1$, $\text{Var}(X_1) = \mathbb{E}(X_1)$ and $\text{Var}(X_2) = \mathbb{E}(X_2)$. For $\mathcal{M}_2$, $\text{Var}(X_1) = \mathbb{E}(X_1)$, while

$$\text{Var}(X_2) = \mathbb{E}[\text{Var}(X_2|X_1)] + \text{Var}[\mathbb{E}(X_2|X_1)] = \mathbb{E}[g_2(X_1)] + \text{Var}[g_2(X_1)] > \mathbb{E}[g_2(X_1)] = \mathbb{E}(X_2),$$

as long as $\text{Var}(g_2(X_1)) > 0$.

Similarly under $\mathcal{M}_3$, $\text{Var}(X_2) = \mathbb{E}(X_2)$ and $\text{Var}(X_1) > \mathbb{E}(X_1)$ as long as $\text{Var}(g_1(X_2)) > 0$. Hence we can identify model $\mathcal{M}_1, \mathcal{M}_2$, and $\mathcal{M}_3$ by testing whether the variance is greater than the expectation or equal to the expectation. With finite sample size $n$, the quantities $\mathbb{E}(\cdot)$ and $\text{Var}(\cdot)$ can be estimated from data and we consider the finite sample setting in Section 4 and 4.2. Now we extend this idea to provide an identifiability condition for general Poisson DAG models.

The key idea to extending identifiability from the bivariate to multivariate scenario involves condition on parents of each node and then testing overdispersion. The general p-variate result is as follows:

**Theorem 3.1.** *Assume that for any $j \in V$, $K \subset Pa(j)$ and $S \subset \{1, 2, .., p\} \setminus K$,*

$$Var(g_j(X_{Pa(j)})|X_S) > 0,$$

*the Poisson DAG model is identifiable.*

We defer the proof to the supplementary material. Once again, the main idea of the proof is overdispersion. To explain the required assumption note that for any $j \in V$ and $S \subset \text{Pa}(j)$, $\text{Var}(X_j|X_S) - \mathbb{E}(X_j|X_S) = \text{Var}(g_j(X_{\mathbf{Pa}(j)})|X_S)$. Note that if $S = \text{Pa}(j)$ or $\{1, ...j-1\}$, $\text{Var}(g_j(X_{\mathbf{Pa}(j)})|X_S) = 0$. Otherwise $\text{Var}(g_j(X_{\mathbf{Pa}(j)})|X_S) > 0$ by our assumption.

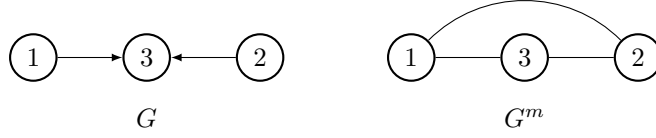

Figure 2: Moralized graph $G^m$ for DAG $G$

## 4 Algorithm

Our algorithm which we call OverDispersion Scoring (ODS) consists of three main steps: 1) estimating a candidate parents set [1, 12, 13] using existing learning undirected graph algorithms; 2) estimating a causal ordering using overdispersion scoring; and 3) estimating directed edges using standard regression algorithms such as Lasso. Steps 3) is a standard problem in which we use off-the-shelf algorithms. Step 1) allows us to reduce both computational and sample complexity by exploiting sparsity of the moralized or undirected graphical model representation of the DAG which we inroduce shortly. Step 2) exploits overdispersion to learn a causal ordering.

An important concept we need to introduce for Step 1) of our algorithm is the *moral* graph or undirected graphical model representation of the DAG (see e.g. [14]). The moralized graph $G^m$ for a DAG $G = (V, E)$ is an undirected graph where $G^m = (V, E_u)$ where $E_u$ includes edge set $E$ without directions plus edges between any nodes that are parents of a common child. Fig. 2 demonstrates concepts of a moralized graph for a simple 3-node example where $E = \{(1,3),(2,3)\}$ for DAG $G$. Note that $1,2$ are parents of a common child 3. Hence $E_u = \{(1,2),(1,3),(2,3)\}$ where the additional edge $(1,2)$ arises from the fact that nodes 1 and 2 are both parents of node 3.

Further, define $\mathcal{N}(j) := \{k \in \{1,2,...,p\} \,|\,(j,k) \text{ or } (k,j) \in E_u\}$ denote the neighborhood set of a node $j$ in the moralized graph $G^m$. Let $\{X^{(i)}\}_{i=1}^n$ denote $n$ samples drawn from the Poisson DAG model $G$. Let $\pi : \{1,2,...,p\} \to \{1,2,...,p\}$ be a bijective function corresponding to a permutation or a causal ordering. We will also use the convenient notation $\widehat{\phantom{x}}$ to denote an estimate based on the data. For ease of notation for any $j \in \{1,2,...p\}$, and $S \subset \{1,2,...,p\}$ let $\mu_{j|S}$ and $\mu_{j|S}(x_S)$ represent $\mathbb{E}(X_j|X_S)$ and $\mathbb{E}(X_j|X_S = x_S)$, respectively. Furthermore let $\sigma^2_{j|S}$ and $\sigma^2_{j|S}(x_S)$ denote $\mathrm{Var}(X_j|X_S)$ and $\mathrm{Var}(X_j|X_S = x_S)$, respectively. We also define $n(x_S) = \sum_{i=1}^n \mathbf{1}(X_S^{(i)} = x_S)$ and $n_S = \sum_{x_S} n(x_S)\mathbf{1}(n(x_S) \geq c_0.n)$ for an arbitrary $c_0 \in (0,1)$.

The computation of the score $\widehat{s}_{jk}$ in Step 2) of our ODS algorithm 1 involves the following equation:

$$\widehat{s}_{jk} = \sum_{x \in \mathcal{X}(\widehat{C}_{jk})} \frac{n(x)}{n_{\widehat{C}_{jk}}} \big(\widehat{\sigma}^2_{j|\widehat{C}_{jk}}(x) - \widehat{\mu}_{j|\widehat{C}_{jk}}(x)\big) \tag{4}$$

where $\widehat{C}_{jk}$ refers to an estimated candidate set of parents specified in Step 2) of our ODS algorithm 1 and $\mathcal{X}(\widehat{C}_{jk}) = \{x \in \{X^{(1)}_{\widehat{C}_{jk}}, X^{(2)}_{\widehat{C}_{jk}}, ..., X^{(n)}_{\widehat{C}_{jk}}\} \mid n(x) \geq c_0.n\}$ so that we ensure we have enough samples for each element we select. In addition, $c_0$ is a tuning parameter of our algorithm that we specify in our main Theorem 4.2 and our numerical experiments.

We can use a number of standard algorithms for Step 1) of our ODS algorithm since it boils down to finding a candidate set of parents. The main purpose of Step 1) is to reduce both computational complexity and the sample complexity by exploiting sparsity in the moralized graph. In Step 1) a candidate set of parents is generated for each node which in principle could be the entire set of nodes. However since Step 2) requires computation of a conditional mean and variance, both the sample complexity and computational complexity depend significantly on the number of variables we condition on as illustrated in Section 4.1 and 4.2. Hence by making the set of candidate parents for each node as small as possible we gain significant computational and statistical improvements by exploiting the graph structure. A similar step is taken in the MMHC [15] and SC algorithms [16]. The way we choose a candidate set of parents is by learning the moralized graph $G^m$ and then using the neighborhood set $\mathcal{N}(j)$ for each $j$. Hence Step 1) reduces to a standard undirected graphical model learning algorithm. A number of choices are available for Step 1) including the neighborhood regression approach of Yang et al. [1] as well as standard DAG learning algorithms which find a candidate parents set such as HITON [13] and MMPC [15].

---

**Algorithm 1: OverDispersion Scoring (ODS)**

---

**input** : $n$ samples from the given Poisson DAG model. $X^{(1)}, ..., X^{(n)} \in \{\{0\} \cup \mathbb{N}\}^p$
**output**: A causal ordering $\widehat{\pi} \in \mathbb{N}^p$ and a graph structure, $\widehat{E} \in \{0, 1\}^{p \times p}$

Step 1: Estimate the undirected edges $\widehat{E}_u$ corresponding to the moralized graph with neighborhood set $\widehat{\mathcal{N}}(j)$;
Step 2: Estimate causal ordering using overdispersion score;
**for** $i \in \{1, 2, ..., p\}$ **do**
   | $\quad \widehat{s}_i = \widehat{\sigma}_i^2 - \widehat{\mu}_i$
**end**
The first element of a causal ordering $\widehat{\pi}_1 = \arg\min_j \widehat{s}_j$;
**for** $j = \{2, 3, ...p - 1\}$ **do**
   **for** $k \in \mathcal{N}(\widehat{\pi}_{j-1}) \cap \{1, 2, ..., p\} \setminus \{\widehat{\pi}_1, ... \widehat{\pi}_{j-1}\}$ **do**
      The candidate parents set $\widehat{C}_{jk} = \widehat{\mathcal{N}}(k) \cap \{\widehat{\pi}_1, \widehat{\pi}_2, ..., \widehat{\pi}_{j-1}\}$;
      Calculate $\widehat{s}_{jk}$ using (4);
   **end**
   The $j^{th}$ element of a causal ordering $\widehat{\pi}_j = \arg\min_k \widehat{s}_{jk}$;
   Step 3: Estimate directed edges toward $\widehat{\pi}_j$, denoted by $\widehat{D}_j$;
**end**
The $p^{th}$ element of the causal ordering $\widehat{\pi}_p = \{1, 2, ..., p\} \setminus \{\widehat{\pi}_1, \widehat{\pi}_2, ..., \widehat{\pi}_{p-1}\}$;
The directed edges toward $\widehat{\pi}_p$, denoted by $\widehat{D}_p = \widehat{\mathcal{N}}(\widehat{\pi}_p)$;
Return the estimated causal ordering $\widehat{\pi} = (\widehat{\pi}_1, \widehat{\pi}_2, ..., \widehat{\pi}_p)$;
Return the estimated edge structure $\widehat{E} = \{\widehat{D}_2, \widehat{D}_3, ..., \widehat{D}_p\}$;

---

Step 2) learns the causal ordering by assigning an overdispersion score for each node. The basic idea is to determine which nodes are overdispersed based on the sample conditional mean and conditional variance. The causal ordering is determined one node at a time by selecting the node with the smallest overdispersion score which is representative of a node that is least likely to be conditionally Poisson and most likely to be marginally Poisson. Finding the causal ordering is usually the most challenging step of DAG learning, since once the causal ordering is learnt, all that remains is to find the edge set for the DAG. Step 3), the final step finds the directed edge set of the DAG $G$ by finding the parent set of each node. Using Steps 1) and 2), finding the parent set of node $j$ boils down to selecting which variables are parents out of the candidate parents of node $j$ generated in Step 1) intersected with all elements before node $j$ of the causal ordering in Step 2). Hence we have $p$ regression variable selection problems which can be performed using GLMLasso [17] as well as standard DAG learning algorithms.

### 4.1 Computational Complexity

Steps 1) and 3) use existing algorithms with known computational complexity. Clearly the computational complexity for Steps 1) and 3) depend on the choice of algorithm. For example, if we use the neighborhood selection GLMLasso algorithm [17] as is used in Yang et al. [1], the worst-case complexity is $O(\min(n, p)np)$ for a single Lasso run but since there are $p$ nodes, the total worst-case complexity is $O(\min(n, p)np^2)$. Similarly if we use GLMLasso for Step 3) the computational complexity is also $O(\min(n, p)np^2)$. As we show in numerical experiments, DAG-based algorithms for Step 1) tend to run more slowly than neighborhood regression based on GLMLasso.

For Step 2) where we estimate the causal ordering has $(p - 1)$ iterations and each iteration has a number of overdispersion scores $\widehat{s}_j$ and $\widehat{s}_{jk}$ computed which is bounded by $O(|K|)$ where $K$ is a set of candidates of each element of a causal ordering, $\mathcal{N}(\widehat{\pi}_{j-1}) \cap \{1, 2, ..., p\} \setminus \{\widehat{\pi}_1, ... \widehat{\pi}_{j-1}\}$, which is also bounded by the maximum degree of the moralized graph $d$. Hence the total number of overdispersion scores that need to be computed is $O(pd)$. Since the time for calculating each overdispersion score which is the difference between a conditional variance and expectation is proportional to $n$, the time complexity is $O(npd)$. In worst case where the degree of the moralized graph is $p$, the computational complexity of Step 2) is $O(np^2)$. As we discussed earlier there is a

significant computational saving by exploiting a sparse moralized graph which is why we perform Step 1) of the algorithm. Hence Steps 1) and 3) are the main computational bottlenecks of our ODS algorithm. The addition of Step 2) which estimates the causal ordering does not significantly add to the computational bottleneck. Consequently our ODS algorithm, which is designed for learning DAGs is almost as computationally efficient as standard methods for learning undirected graphical models.

## 4.2 Statistical Guarantees

In this section, we show consistency of recovering a valid causal ordering recovery of our ODS algorithm under suitable regularity conditions. We begin by stating the assumptions we impose on the functions $g_j(.)$.

**Assumption 4.1.**

(A1) For all $j \in V$, $K \subset \text{Pa}(j)$ and all $S \subset \{1, 2.., p\} \setminus K$, there exists an $m > 0$ such that $\text{Var}(g_j(X_{\text{Pa}(j)})|X_S) > m$.

(A2) For all $j \in V$, there exists an $M < \infty$ such that $\mathbb{E}[\exp(g_j(X_{\text{Pa}(j)}))] < M$.

(A1) is a stronger version of the identifiability assumption in 3.1 $\text{Var}(g_j(X_{\text{Pa}(j)})|X_S) > 0$ where since we are in the finite sample setting, we need the conditional variance to be lower bounded by a constant bounded away from 0. (A2) is a condition on the tail behavior of $g_j(\text{Pa}(j))$ for controlling tails of the score $\hat{s}_{jk}$ in Step 2 of our ODS algorithm. To take a concrete example for which (A1) and (A2) are satisfied, it is straightforward to show that the GLM DAG model (2) with non-positive values of $\{\theta_{kj}\}$ satisfies both (A1) and (A2). The non-positivity constraint on the $\theta$'s is sufficient but not necessary and ensures that the parameters do not grow too large.

Now we present the main result under Assumptions (A1) and (A2). For general DAGs, the true causal ordering $\pi^*$ is not unique. Therefore let $\mathcal{E}(\pi^*)$ denote all the causal orderings that are consistent with the true DAG $G^*$. Further recall that $d$ denotes the maximum degree of the moralized graph $G_m^*$.

**Theorem 4.2** (Recovery of a causal ordering). *Consider a Poisson DAG model as specified in* (1)*, with a set of true causal orderings $\mathcal{E}(\pi^*)$ and the rate function $g_j(.)$ satisfies assumptions 4.1. If the sample size threshold parameter $c_0 \leq n^{-1/(5+d)}$, then there exist positive constants, $C_1, C_2, C_3$ such that*

$$\mathbb{P}(\hat{\pi} \notin \mathcal{E}(\pi^*)) \leq C_1 exp(-C_2 n^{1/(5+d)} + C_3 \log \max\{n, p\}).$$

We defer the proof to the supplementary material. The main idea behind the proof uses the overdispersion property exploited in Theorem 3.1 in combination with concentration bounds that exploit Assumption (A2). Note once again that the maximum degree $d$ of the undirected graph plays an important role in the sample complexity which is why Step 1) is so important. This is because the size of the conditioning set depends on the degree of the moralized graph $d$. Hence $d$ plays an important role in both the sample complexity and computational complexity.

Theorem 4.2 can be used in combination with sample complexity guarantees for Steps 1) and 3) of our ODS algorithm to prove that our output DAG $\hat{G}$ is the true DAG $G^*$ with high probability. Sample complexity guarantees for Steps 1) and 3) depend on the choice of algorithm but for neighborhood regression based on the GLMLasso, provided $n = \Omega(d \log p)$, Steps 1) and 3) should be consistent.

For Theorem 4.2 if the triple $(n, d, p)$ satisfies $n = \Omega((\log p)^{5+d})$, then our ODS algorithm recovers the true DAG. Hence if the moralized graph is sparse, ODS recovers the true DAG in the high-dimensional $p > n$ setting. DAG learning algorithms that apply to the high-dimensional setting are not common since they typically rely on faithfulness or similar assumptions or other restrictive conditions that are not satisfied in the $p > n$ setting. Note that if the DAG is not sparse and $d = \Omega(p)$, our sample complexity is extremely large when $p$ is large. This makes intuitive sense since if the number of candidate parents is large, we would need to condition on a large set of variables which is very sample-intensive. Our sample complexity is certainly not optimal since the choice of tuning parameter $c_0 \leq n^{-1/(5+d)}$. Determining optimal sample complexity remains an open question.

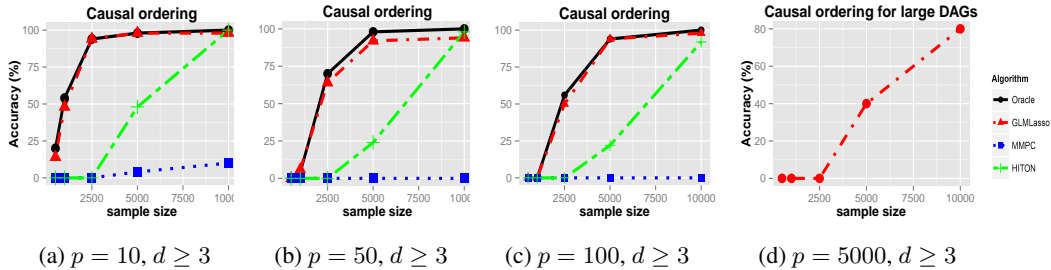

|(a) $p = 10$, $d \geq 3$ | (b) $p = 50$, $d \geq 3$ | (c) $p = 100$, $d \geq 3$ | (d) $p = 5000$, $d \geq 3$ |

Figure 3: Accuracy rates of successful recovery for a causal ordering via our ODS algorithm using different base algorithms

The larger sample complexity of our ODS algorithm relative to undirected graphical models learning is mainly due to the fact that DAG learning is an intrinsically harder problem than undirected graph learning when the causal ordering is unknown. Furthermore note that Theorem 4.2 does not require any additional identifiability assumptions such as faithfulness which severely increases the sample complexity for large-scale DAGs [6].

# 5 Numerical Experiments

In this section, we support our theoretical results with numerical experiments and show that our ODS algorithm performs favorably compared to state-of-the-art DAG learning methods. The simulation study was conducted using 50 realizations of a $p$-node random Poisson DAG that was generated as follows. The $g_j(.)$ functions for the general Poisson DAG model (1) was chosen using the standard GLM link function (i.e. $g_j(X_{\mathbf{Pa}(j)}) = \exp(\theta_j + \sum_{k \in \mathbf{Pa}(j)} \theta_{jk} X_k)$) resulting in the GLM DAG model (2). We experimented with other choices of $g_j(.)$ but only present results for the GLM DAG model (2). Note that our ODS algorithm works well as long as Assumption 4.1 is satisfied regardless of choices of $g_j(.)$. In all results presented $(\theta_{jk})$ parameters were chosen uniformly at random in the range $\theta_{jk} \in [-1, -0.7]$ although any values far from zero and satisfying the assumption 4.1 work well. In fact, smaller values of $\theta_{jk}$ are more favorable to our ODS algorithm than state-of-the-art DAG learning methods because of weak dependency between nodes. DAGs are generated randomly with a fixed unique causal ordering $\{1, 2..., p\}$ with edges randomly generated while respecting desired maximum degree constraints for the DAG. In our experiments, we always set the thresholding constant $c_0 = 0.005$ although any value below 0.01 seems to work well.

In Fig. 3, we plot the proportion of simulations in which our ODS algorithm recovers the correct causal ordering in order to validate Theorem 4.2. All graphs in Fig. 3 have exactly 2 parents for each node and we plot how the accuracy in recovering the true $\pi^*$ varies as a function of $n$ for $n \in \{500, 1000, 2500, 5000, 10000\}$ and for different node sizes (a) $p = 10$, (b) $p = 50$, (c) $p = 100$, and (d) $p = 5000$. As we can see, even when $p = 5000$, our ODS algorithm recovers the true causal ordering about 40% of the time even when $n$ is approximately 5000 and for smaller DAGs accuracy is 100%. In each sub-figure, 3 different algorithms are used for Step 1): GLMLasso [17] where we choose $\lambda = 0.1$; MMPC [15] with $\alpha = 0.005$; and HITON [13] again with $\alpha = 0.005$ and an oracle where the edges for the true moralized graph is used. As Fig. 3 shows, the GLMLasso seems to be the best performing algorithm in terms of recovery so we use the GLMLasso for Steps 1) and 3) for the remaining figures. GLMLasso was also the only algorithm that scaled to the $p = 5000$ setting. However, it should be pointed out that GLMLasso is not necessarily consistent and it is highly depending on the choice of $g_j(.)$. Recall that the degree $d$ refers to the maximum degree of the moralized DAG.

Fig. 4 provides a comparison of how our ODS algorithm performs in terms of Hamming distance compared to the state-of-the-art PC [3], MMHC [15], GES [18], and SC [16] algorithms. For the PC, MMHC and SC algorithms, we use $\alpha = 0.005$ while for the GES algorithm we use the mBDe [19] (modified Bayesian Dirichlet equivalent) score since it performs better than other score choices. We consider node sizes of $p = 10$ in (a) and (b) and $p = 100$ in (c) and (d) since many of these algorithms do not easily scale to larger node sizes. We consider two Hamming distance measures: in (a) and (c), we only measure the Hamming distance to the skeleton of the true DAG, which is the set of edges of the DAG without directions; for (b) and (d) we measure the Hamming distance for

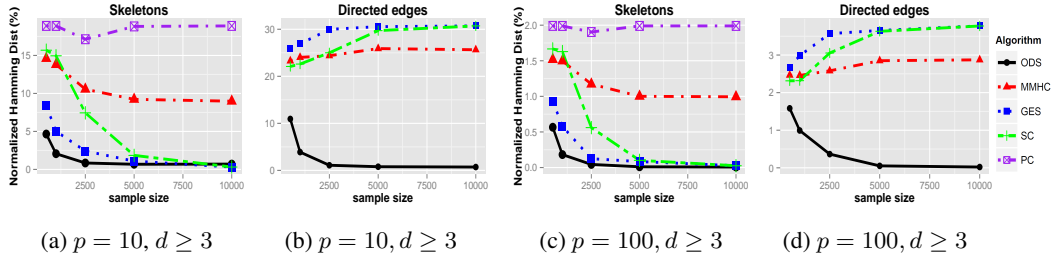

(a) $p = 10, d \geq 3$  (b) $p = 10, d \geq 3$  (c) $p = 100, d \geq 3$  (d) $p = 100, d \geq 3$

Figure 4: Comparison of our ODS algorithm (black) and PC, GES, MMHC, SC algorithms in terms of Hamming distance to skeletons and directed edges.

the edges with directions. The reason we consider the skeleton is because the PC does not recover all directions of the DAG. We normalize the Hamming distance by dividing by the total number of edges $\binom{p}{2}$ and $p(p-1)$, respectively so that the overall score is a percentage. As we can see our ODS algorithm significantly out-performs the other algorithms. We can also see that as the sample size $n$ grows, our algorithm recovers the true DAG which is consistent with our theoretical results. It must be pointed out that the choice of DAG model is suited to our ODS algorithm while these state-of-the-art algorithms apply to more general classes of DAG models.

Now we consider the statistical performance for large-scale DAGs. Fig. 5 plots the statistical performance of ODS for large-scale DAGs in terms of (a) recovering the causal ordering; (b) Hamming distance to the true skeleton; (c) Hamming distance to the true DAG with directions. All graphs in Fig. 5 have exactly 2 parents for each node and accuracy varies as a function of $n$ for $n \in \{500, 1000, 2500, 5000, 10000\}$ and for different node sizes $p = \{1000, 2500, 5000\}$. Fig. 5 shows that our ODS algorithm accurately recovers the causal ordering and true DAG models even in high dimensional setting, supporting our theoretical results 4.2.

Fig. 6 shows run-time of our ODS algorithm. We measure the running time (a) by varying node size $p$ from 10 to 125 with the fixed $n = 100$ and 2 parents; (b) sample size $n$ from 100 to 2500 with the fixed $p = 20$ and 2 parents; (c) the number of parents of each node $|\text{Pa}|$ from 1 to 5 with the fixed $n = 5000$ and $p = 20$. Fig. 6 (a) and (b) support the section 4.1 where the time complexity of our ODS algorithm is at most $O(np^2)$. Fig. 6 (c) shows running time is proportional to a parents size which is a minimum degree of a graph. It agrees with the time complexity of Step 2) of our ODS algorithm is $O(npd)$. We can also see that the GLMLasso has the fastest run-time amongst all algorithms that determine the candidate parent set.

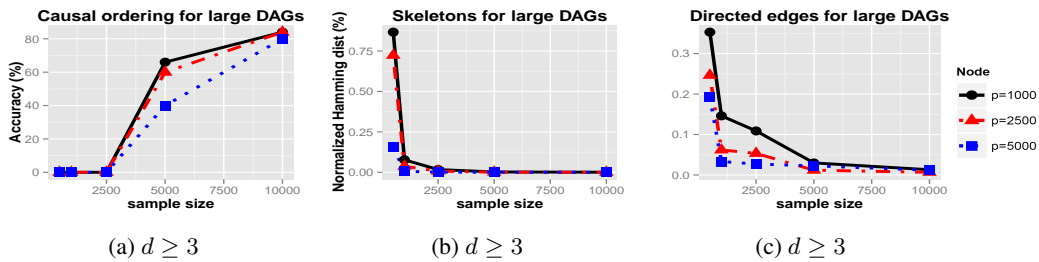

(a) $d \geq 3$  (b) $d \geq 3$  (c) $d \geq 3$

Figure 5: Performance of our ODS algorithm for large-scale DAGs with $p = 1000, 2500, 5000$

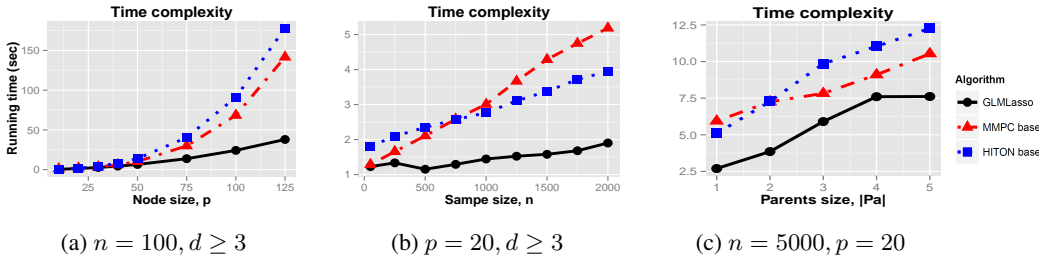

(a) $n = 100, d \geq 3$  (b) $p = 20, d \geq 3$  (c) $n = 5000, p = 20$

Figure 6: Time complexity of our ODS algorithm with respect to node size $p$, sample size $n$, and parents size $|\text{Pa}|$

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
