[Supplementary Material]

# Supplementary Material

**Gunwoong Park**
Department of Statistics
University of Wisconsin-Madison
Madison, WI 53706
parkg@stat.wisc.edu

**Garvesh Raskutti**
Department of Statistics
Department of Computer Science
Wisconsin Institute for Discovery, Optimization Group
University of Wisconsin-Madison
Madison, WI 53706
raskutti@cs.wisc.edu

## 1 Proof of Theorem 3.1

*Proof.* We prove it by induction that requires $p$ steps to find a causal ordering that is consistent with the DAG. Without loss of generality, assume that one of the true causal ordering $\pi^*$ is $\{1, 2, ...p\}$. For ease of notation, let $\mathcal{F}_s = \{X_1, X_2, \cdots, X_s\}$. Let $k = 1$ be the first step:

$$\text{Var}(X_j) = \mathbb{E}(\text{Var}[X_j|\mathcal{F}_{j-1}]) + \text{Var}(\mathbb{E}[X_j|\mathcal{F}_{j-1}]),$$

where the outer expectation and variance is taken over $X_1, X_2, ..., X_{j-1}$. Since the conditional distribution $X_j|\mathcal{F}_{j-1} \sim \text{Poisson}(g_j(X_{\mathbf{Pa}(j)}))$, we have $\text{Var}[X_j|\mathcal{F}_{j-1}] = \mathbb{E}[X_j|\mathcal{F}_{j-1}] = g_j(X_{\mathbf{Pa}(j)})$. Hence,

$$\text{Var}(X_j) = \mathbb{E}(\mathbb{E}[X_j|\mathcal{F}_{j-1}]) + \text{Var}(g_j(X_{\mathbf{Pa}(j)}))$$
$$= \mathbb{E}(X_j) + \text{Var}(g_j(X_{\mathbf{Pa}(j)})),$$

yielding that

$$\text{Var}(X_j) - \mathbb{E}(X_j) = \text{Var}(g_j(X_{\mathbf{Pa}(j)})).$$

Clearly, if $\text{Pa}(j)$ is empty, meaning the node is the first component of the causal ordering, $\text{Var}(g_j(X_{\mathbf{Pa}(j)})) = 0$. Otherwise, $\text{Var}(g_j(X_{\mathbf{Pa}(j)})) > 0$ by the assumption. Hence for any node that can not be the first in the ordering, $\text{Var}(X_j) - \mathbb{E}(X_j) > 0$. Hence we pick any node $X_k$ such that $\text{Var}(X_k) - \mathbb{E}(X_k) = 0$ as being the first element of the causal ordering and $X_1$ satisfies the above equation.

For $k = m$, assume $X_1, X_2, ..., X_m$ is a valid causal ordering for the first $m$ nodes. Now we consider

$$\text{Var}(X_j|\mathcal{F}_m) = \mathbb{E}(\text{Var}[X_j|\mathcal{F}_{j-1}]|\mathcal{F}_m) + \text{Var}(\mathbb{E}[X_j|\mathcal{F}_{j-1}]|\mathcal{F}_m),$$

for $j = m + 1, m + 2, ..., p$, where the expectation and variance are taken over the variables $X_1, X_2, ..., X_m$. Again, for any $j = m + 1, m + 2, ..., p$, we have $\text{Var}[X_j|\mathcal{F}_{j-1}] = \mathbb{E}[X_j|\mathcal{F}_{j-1}] = g_j(X_{\mathbf{Pa}(j)})$. Further, since $X_1, X_2, ..., X_m$ is a valid causal ordering for the first $m$ nodes,

$$\text{Var}(X_j|\mathcal{F}_m) = \mathbb{E}(\mathbb{E}[X_j|\mathcal{F}_{j-1}]|\mathcal{F}_m) + \text{Var}(\mathbb{E}(X_j|\mathcal{F}_{j-1})|\mathcal{F}_m)$$
$$= \mathbb{E}(X_j|\mathcal{F}_m) + \text{Var}(g_j(X_{\mathbf{Pa}(j)})|\mathcal{F}_m).$$

Hence, following on similar lines,

$$\text{Var}(X_j|\mathcal{F}_m) - \mathbb{E}(X_j|\mathcal{F}_m) = \text{Var}[g_j(X_{\mathbf{Pa}(j)})|\mathcal{F}_m].$$

Hence if $\text{Pa}(j) \setminus \{1, 2, ..., m\}$ is empty, $\text{Var}(g_j(X_{\mathbf{Pa}(j)})|\mathcal{F}_m) = 0$ and $\text{Var}(X_j|\mathcal{F}_m) - \mathbb{E}(X_j|\mathcal{F}_m) = 0$. Any such node can be next on the causal ordering and $X_m$ holds the above property. On the other hand, for any node in which $\text{Pa}(j) \setminus \{1, 2, ..., m\}$ is non-empty $\text{Var}(X_j|\mathcal{F}_m) - \mathbb{E}(X_j|\mathcal{F}_m) > 0$ which excludes it from being next in the causal ordering. Hence $X_1, X_2, ..., X_{m+1}$ is a valid causal ordering for the first $m + 1$ nodes. This completes the proof by induction. $\qquad\square$

## 2 Proof of Theorem 4.2

*Proof.* Let $X^{(i)} = (X_1^{(i)}, ..., X_p^{(i)})$ be the i.i.d $n$ samples from the given DAG model. Let $\pi^*$ be a true causal ordering and $\hat{\pi}$ be the estimated causal ordering. Without loss of generality, assume that the true causal ordering $\pi^*$ is $\{1, 2, ...p\}$. For an arbitrary permutation or causal ordering $\pi$, let $\pi_j$ represent its $j^{th}$ element.

Let $E_u$ denote the set of undirected edges corresponding to the *moralized* graph (i.e. the directed edges without directions and edges between nodes with common children). Recall the definitions $\mathcal{N}(j) := \{k \in \{1, 2, ..., p\} \,|\, (j, k) \in E_u\}$ denote the neighborhood set of $j$ in the moralized graph and $K(j) = \{k | k \in \mathcal{N}(j-1) \cap \{j, ..., p\}\}$ denote a candidate set for $\pi_j$ and $C_{jk} = \mathcal{N}(k) \cap \{\pi_1, \pi_2, ..., \pi_{j-1}\}$ which is the intersection of the neighbors of $k$ with $\{1, 2, ..., j-1\}$.

Recall that for ease of notation for any $j \in \{1, 2, ...p\}$, and $S \subset \{1, 2, ..., p\}$ let $\mu_{j|S}$ and represent $\mathbb{E}[X_j | X_S]$ and $\sigma_{j|S}^2 = \text{Var}(X_j | X_S)$. Also, denote $\mu_{j|S}(x_S)$ and represent $\mathbb{E}[X_j | X_S = x_S]$ and $\sigma_{j|S}^2(x_S) = \text{Var}(X_j | X_S = x_S)$. Let $n_S(x_S) = \sum_{i=1}^n \mathbf{1}(X_S^{(i)} = x_S)$ and $n_S = \sum_{x_S} n(x_S) \mathbf{1}(n(x_S) \geq c_0.n)$ for an arbitrary $c_0 \in (0, 1)$.

The overdispersion score of $k \in K(j)$ for the $j^{th}$ component of the causal ordering, defined in the second step of our ODS algorithm only considers elements of $\mathcal{X}(\hat{C}_{jk}) = \{x \in \{X_{\hat{C}_{jk}}^{(1)}, X_{\hat{C}_{jk}}^{(2)}, ..., X_{\hat{C}_{jk}}^{(n)}\} \,|\, n(x) \geq c_0.n\}$ so we only count up elements that occur sufficiently frequently.

According to the ODS algorithm, the truncated sample conditional expectation and variance of $X_j$ given $X_S = x$ for $j \in \{1, 2, ...p\}$ and any subset $S \subset \{1, 2, ...p\} \setminus \{j\}$ be following: for $x \in \mathcal{X}(S)$,

$$\widehat{\mu}_{j|S}(x) = \frac{1}{n_S(x)} \sum_{i=1}^n X_j^{(i)} \mathbf{1}(X_S^{(i)} = x)$$

$$\widehat{\sigma}_{j|S}^2(x) = \frac{1}{n_S(x) - 1} \sum_{i=1}^n (X_j^{(i)} - \widehat{\mu}_{j|S}(x))^2 \mathbf{1}(X_S^{(i)} = x)$$

The overdispersion score of $k \in K(j)$ for the $j^{th}$ element of the causal ordering is for $x \in \mathcal{X}(C_{jk})$,

$$\widehat{s}_{jk}(x) = \widehat{\sigma}_{k|\hat{C}_{jk}}^2(x) - \widehat{\mu}_{k|\hat{C}_{jk}}(x)$$

$$\widehat{s}_{jk} = \widehat{\mathbb{E}}_{\hat{C}_{jk}}(\widehat{s}_{jk}(x)) = \sum_{x \in \mathcal{X}(jk)} \frac{n_{\hat{C}_{jk}}(x)}{n_{\hat{C}_{jk}}} \widehat{s}_{jk}(x).$$

And the correct overdispersion score is

$$s_{jk}^* = \mathbb{E}_{C_{jk}}[\sigma_{k|C_{jk}}^2 - \mu_{k|C_{jk}}] = \mathbb{E}_{C_{jk}}[\text{Var}(g_k(\text{Pa}(k))|C_{jk})].$$

Let us define some events for the proof and $d$ denote the maximum degree of the moralized graph. For any $j \in \{1, 2, ..., p\}$ and $k \in K(j)$,

$$\xi_1 = \{\max_{j,k} |\widehat{s}_{jk} - s_{jk}^*| < m/2\}$$

$$\xi_2 = \{\max_k \max_{i=1,...,n} X_k^{(i)} < n^{\frac{1}{5+d}}\}$$

We prove it by induction that requires $p$ steps to recover a causal ordering that is consistent with the Poisson DAG. Without loss of generality, assume that the true causal ordering $\pi^*$ is $\{1, 2, ...p\}$. For the first step $j = 1$, a set of candidate element of $\pi_1$ is $K(1) = \{1, 2, ...., p\}$ and a candidate parent set of each node $C_{1k} = \emptyset$ for all $k \in K(1)$.

$$P(\widehat{\pi}_1 \neq \pi_1^*) = P\left(\text{exists at least one } k \in K(1) \setminus \{1\} \text{ s.t. } \widehat{s}_{11} > \widehat{s}_{1k}\right)$$

$$\leq |K(1)| \max_{k \in K(1) \setminus \{1\}} \left\{ P\left(s_{11}^* + \frac{m}{2} > s_{1k}^* - \frac{m}{2} | \xi_1\right) + P(\xi_1^c | \xi_2) + P(\xi_2^c) \right\}$$

$$\leq p \max_{k \in K(1) \setminus \{1\}} \left\{ P\left(m > s_{1k}^* | \xi_1\right) + P(\xi_1^c | \xi_2) + P(\xi_2^c) \right\}$$

By Assumption (A1), $s^*_{1k} > m$ and we will represent some Propositions that respectively control $P(\xi^c_1|\xi_2)$ and $P(\xi^c_2)$.

For the $j-1$ step, assume $(\widehat{\pi}_1, \widehat{\pi}_2, ..., \widehat{\pi}_{j-1})$ is a valid ordering for the first $j-1$ nodes. Note that with the correct $\mathcal{N}(j)$, $\widehat{C}_{jk} = C_{jk}$. Now, we consider $\pi^*_j$. The probability of a false recovery of $\pi^*_j$ given the true undirected edges of the moralized graph and the true causal ordering before $j$ is following:

$$
\begin{aligned}
&\mathrm{P}(\widehat{\pi}_j \neq \pi^*_j | \widehat{\pi}_1 = \pi^*_1, ..., \widehat{\pi}_{j-1} = \pi^*_{j-1}) \\
&= P\big(\text{exists at least one } k \in K(j) \setminus \{j\} \text{ s.t. } \widehat{s}_{jj} > \widehat{s}_{jk}\big) \\
&\leq |K(j)| \max_{k \in K(j)\setminus\{j\}} \big\{ P\big(\widehat{s}_{jj} + m/2 > s^*_{jk} - m/2 | \xi_1\big) + P(\xi^c_1|\xi_2) + P(\xi^c_2) \big\} \\
&\leq |K(j)| \max_{k \in K(j)\setminus\{j\}} \big\{ P\big(m > s^*_{jk} | \xi_1\big) + P(\xi^c_1|\xi_2) + P(\xi^c_2) \big\}
\end{aligned}
$$

By Assumption (A1), $s^*_{jk} > m$ and we represent some Propositions that respectively control $P(\xi^c_1|\xi_2)$ and $P(\xi^c_2)$. Furthermore we also show a condition on $c_0$.

**Proposition 2.1.** *For all $j \in \{1, 2, ..., p\}, k \in K(j)$, $c_0 \leq n^{-\frac{d}{5+d}}$ given $\xi_2$ is a sufficient that a candidate parents set $\mathcal{X}(C_{jk})$ is not empty*

**Proposition 2.2.**

$$
P(\xi^c_1|\xi_2) \leq 2p^2 n^{\frac{d}{5+d}} \big\{ exp\big(-\frac{m^2 n^{1/(5+d)}}{18}\big) + exp\big(-\frac{m^2 n^{1/(5+d)}}{9}\big) + exp\big(-\frac{m^2 n^{3/(5+d)}}{9}\big) \big\},
$$

*where $m$ is the constant in Assumption (A1).*

**Proposition 2.3.**

$$
P(\xi^c_2) \leq npM exp\big(-n^{1/(5+d)} \log 2\big)
$$

*where $M$ is the constant in Assumption (A2).*

Hence for any $j \in \{1, 2, ...p\}$ with $c_0 = n^{-\frac{d}{5+d}}$,

$$
\begin{aligned}
&\mathrm{P}(\widehat{\pi}_j \neq \pi^*_j | \widehat{\pi}_1 = \pi^*_1, ..., \widehat{\pi}_{j-1} = \pi^*_{j-1}) \\
&\leq p \max_{k \in K(j)\setminus\{j\}} \big\{ P\big(m > s^*_{jk} | \xi_1\big) + P(\xi^c_1|\xi_2) + P(\xi^c_2) \big\} \\
&\leq 2p^3 n^{\frac{d}{5+d}} \big\{ \exp\big(-\frac{m^2 n^{1/(5+d)}}{18}\big) + \exp\big(-\frac{m^2 n^{1/(5+d)}}{9}\big) + \exp\big(-\frac{m^2 n^{3/(5+d)}}{9}\big) \big\} \\
&+ np^2 M\exp\big(-n^{1/(5+d)} \log 2\big)
\end{aligned}
\tag{1}
$$

By using the above probability bound (1),

$$
\begin{aligned}
P(\widehat{\pi} \neq \pi^*) &\overset{(E_1)}{\leq} P(\widehat{\pi}_1 \neq \pi^*_1) + ... + P(\widehat{\pi}_{p-1} \neq \pi^*_{p-1} | \widehat{\pi}_1 = \pi^*_1, ..., \widehat{\pi}_{p-2} = \pi^*_{p-2}) \\
&\overset{(E_2)}{\leq} 2p^4 n^{\frac{d}{5+d}} \big\{ \exp\big(-\frac{m^2 n^{1/(5+d)}}{18}\big) + \exp\big(-\frac{m^2 n^{1/(5+d)}}{9}\big) + \exp\big(-\frac{m^2 n^{3/(5+d)}}{9}\big) \big\} \\
&+ np^3 M\exp\big(-n^{1/(5+d)} \log 2\big)
\end{aligned}
$$

The first inequality $(E_1)$ is followed from $P(A \cup B) = P(A) + P(B \cap A^c) = P(A) + P(B|A^c)P(A^c) \leq P(A) + P(B|A^c)$ for some events $A, B$. And $(E_2)$ is directly from (1).

Hence, there exists some positive constants $C_1, C_2, C_3 > 0$ such that

$$
P(\widehat{\pi} \neq \pi^*) \leq C_1 \exp\big(-C_2 n^{1/(5+d)} + C_3 \log \max\{p, n\}\big)
$$

$\square$

### 2.0.1 Proof of Proposition 2.1

*Proof.* Let $|X_S|$ denote the cardinality of a set $\{X_S^{(1)}, X_S^{(2)}, ..., X_S^{(n)}\}$ and $|\mathcal{X}(S)|$ denote the cardinality of a set $\mathcal{X}(S)$. In worst case where $|\mathcal{X}(S)| = 1$, for all $x \in \{X_S^{(1)}, X_S^{(2)}, ..., X_S^{(n)}\}$, $n_S(x) = c_0.n - 1$ except for only one component $y \in \mathcal{X}(S)$. In this case, the sample size $n = n_S(y) + (|X_S| - 1)(c_0.n - 1)$. A simple calculation yields that

$$n_S(y) = n - (|X_S| - 1)(c_0.n - 1) = n - c_0.n|X_S| + c_0.n + |X_S| - 1.$$

Hence $c_0.n \le n_S(y)$ is equivalent to $c_0 \le \frac{n+|X_S|-1}{n.|X_S|}$. Since $\frac{1}{|X_S|} \le \frac{n+|X_S|-1}{n|X_S|}$, if $c_0 \le \frac{1}{|X_S|}$ there exists at least one component $y \in \mathcal{X}(S)$. In addition under the event $\xi_2$, $|X_S| \le n^{\frac{d}{5+d}}$ which is all possible combinations. Hence if $c_0 \le n^{-\frac{d}{5+d}}$, $|\mathcal{X}(S)| \ne 0$. $\square$

### 2.0.2 Proof of Proposition 2.2

*Proof.* This problem is reduced to the consistency rate of a sample conditional mean and conditional variance. For ease of notation, let $n_{jk} = n_{C_{jk}}$ and $n_{jk}(x) = n_{C_{jk}}(x)$. Suppose that $c_0 = n^{-\frac{d}{5+d}}$. Then for any $j \in \{1, 2, ..., p\}$ and $k \in K(j)$,

$$P(\xi_1^c | \xi_2) \le p^2 \max_{j,k} P(|\widehat{s}_{jk} - s_{jk}^*| > \frac{m}{2} | \xi_2)$$

$$\le p^2 \max_{j,k} P(\sum_{x \in \mathcal{X}(C_{jk})} \frac{n_{jk}(x)}{n_{jk}} |\widehat{s}_{jk}(x) - s_{jk}^*(x)| > \frac{m}{2} | \xi_2)$$

$$\overset{(E_1)}{\le} p^2 \max_{j,k} \sum_{x \in \mathcal{X}(C_{jk})} P(|\widehat{s}_{jk}(x) - s_{jk}^*(x)| > \frac{m}{2} \frac{n_{jk}}{n_{jk}(x)} | \xi_2)$$

$$\overset{(E_2)}{\le} p^2 \max_{j,k} |\mathcal{X}(C_{jk})| \max_{x \in \mathcal{X}(C_{jk})} P(|\widehat{s}_{jk}(x) - s_{jk}^*(x)| > \frac{m}{2} | \xi_2)$$

$$\overset{(E_3)}{\le} p^2 n^{\frac{d}{5+d}} \max_{j,k,x} P(|(\widehat{\sigma}_{k|C_{jk}}^2(x) - \widehat{\mu}_{k|C_{jk}}(x)) - (\sigma_{k|C_{jk}}^2(x) - \mu_{k|C_{jk}}(x))| > \frac{m}{2} | \xi_2)$$

$$\le p^2 n^{\frac{d}{5+d}} \max_{j,k,x} \{P(|\widehat{\sigma}_{k|C_{jk}}^2(x) - \sigma_{j|C_{jk}}^2(x)| > \frac{m}{3} | \xi_2) + P(|\widehat{\mu}_{k|C_{jk}}(x) - \mu_{k|C_{jk}}(x)| > \frac{m}{6} | \xi_2)\}$$

$$\overset{(E_4)}{\le} 2p^2 n^{\frac{d}{5+d}} \max_{j,k,x} \{\exp(-\frac{m^2 n_{jk}(x)}{18 n^{4/(5+d)}}) + \exp(-\frac{m^2 n_{jk}(x)}{9 n^{4/(5+d)}}) + \exp(-\frac{m^2 n_{jk}(x)}{9 n^{2/(5+d)}})\}$$

$$\overset{(E_5)}{\le} 2p^2 n^{\frac{d}{5+d}} \max_{j,k,x} \{\exp(-\frac{m^2 n^{1/(5+d)}}{18}) + \exp(-\frac{m^2 n^{1/(5+d)}}{9}) + \exp(-\frac{m^2 n^{3/(5+d)}}{9})\}$$

$$= 2p^2 n^{\frac{d}{5+d}} \{\exp(-\frac{m^2 n^{1/(5+d)}}{18}) + \exp(-\frac{m^2 n^{1/(5+d)}}{9}) + \exp(-\frac{m^2 n^{3/(5+d)}}{9})\}.$$

$(E_1)$ is followed from that $P(\sum_i \omega_i X_i > \delta) \le \sum_i P(X_i > \delta/\omega_i)$, and $(E_2)$ is from $\frac{n_{jk}(x)}{n_{jk}} < 1$. Since $n_{jk}(x) \ge c_0.n$ for all $x \in \mathcal{X}(C_{jk})$, $|\mathcal{X}(C_{jk})| \le 1/c_0$ hence $(E_3)$ and $(E_5)$ hold. Moreover, $(E_4)$ is followed from the Hoeffding's inequality (Theorem 2 [1]) since samples are independent and bounded above $n^{1/(5+d)}$ given $\xi_2$. $\square$

### 2.0.3  Proof of Proposition 2.3

*Proof.* For any $j \in \{1, 2, ..., p\}$, the conditional distribution of $X_j$ given $X_{\mathrm{pa}(j)}$ is Poisson with rate parameter $g_j(\mathrm{Pa}(j))$. Hence for $k \in K(j)$,

$$
\begin{aligned}
P(\xi_2^c) &= P\big(\max_{k \in K(j)} \max_{i=1,...,n} X_k^{(i)} > n^{1/(5+d)}\big) \\[4pt]
&\overset{(E_1)}{\leq} np \max_{k \in K(j)} \max_{i=1,...,n} P(|X_k^{(i)}| > n^{1/(5+d)}) \\[4pt]
&\overset{(E_2)}{\leq} np \max_{k \in K(j)} \max_{i=1,...,n} \mathbb{E}_{\mathrm{pa}(k)}\big[\exp\big(-n^{1/(5+d)} \log 2 + g_k(\mathrm{pa}(k))\big)\big] \\[4pt]
&\overset{(E_3)}{\leq} np \max_{k \in K(j)} \max_{i=1,...,n} M \exp(-n^{1/(5+d)} \log 2) \\[4pt]
&= npM \exp\big(-n^{1/(5+d)} \log 2\big).
\end{aligned}
$$

$(E_1)$ is followed from the union bound and $|K(j)| < p$. $(E_2)$ is from the moment generating function of Poisson distribution with $t = \log 2$. And, $(E_3)$ is from Assumption (A2). $\square$

## References

[1] W. Hoeffding, "Probability inequalities for sums of bounded random variables," *Journal of the American statistical association*, vol. 58, no. 301, pp. 13–30, 1963.