[Reviews · NeurIPS 2015]

Submitted by Assigned_Reviewer_1

The paper introduces the class of Poisson DAG models, directed graphical models where each node is conditionally Poisson on its parents, e.g. through a GLM model. This particular subclass of models may be of use in applications where count data is frequent, e.g. bioinformatics (although the authors don't really do a good job of motivating the approach). The authors prove using an intuitive notion of overdispersion that the causal ordering in these models is identifiable, and then provide an algorithm which shows impressive performance in the simulations results, and with theoretical guarantees (although I wonder how tight the bound in Thm 4.2 is, it looks to me like it just tells me that a probability is smaller than 1 in many cases). My reservations on the paper are the following: - it is not an easy paper to read, motivation is minimal, discussion is absent. The notation around eqn (4) beat me, I could not find the definition of C_{jk} and anyway I don't get what it means x\in X_{C_{jk}} when x should then be an integer number as it is a value taken by a Poisson variable. Also, I appreciate that the authors focus on the novel step 2 of the algorithm, but a minimal discussion of the other steps would not go amiss. - evaluation, this is impressive but it does rely heavily on a model correctness assumption. I would have liked to see how the algorithm behaves under model mismatch (e.g. you tell it maximum in-degree 2 but actually the data comes from an in-degree 3 network) - the paper clearly has its strengths in the theoretical/ algorithmic part, however an example that had at least some motivation in reality would be good. Having said these things, I think the idea is novel and interesting and this could be a quality contribution.
Summary: A paper presenting interesting novel theoretical results plus an algorithm for learning DAG models with conditionally Poisson variables. I found it worthwhile even though there could be improvements.

Submitted by Assigned_Reviewer_2

This paper extends graphical models determined by acyclic digraphs (DAGs) to count data by using Poisson models. The basic assumption that variables that are conditionally Poisson are overdispersed relative to variables that are marginally Poisson is not connected to the notion of conditional (in)dependence, which is IMO a basic feature to show in any new formulation of a graphical model. In the numerical experiments, the authors compare their approach to other methods that, to my understanding, they are not prepared to work with count data. So, it is necessary that the authors give more detail about how those methods were run on those data, for instance, were the count data log-transformed?
Summary: This paper presents an algorithm to learn DAG models from count data using Poisson distributions. The key methodological contribution is to assume that variables that are conditionally Poisson are overdispersed relative to variables that are marginally Poisson. While the problem and the approach based on Poisson models is appealing, the paper IMO seems unfinished.

Submitted by Assigned_Reviewer_3

The paper considers the inference of the structure of the DAG underlying a Poisson directed graphical model. The proposed strategy relies on the fact that nodes with parents are conditionally Poisson, whereas nodes without parents are marginally Poisson. The paper aims at distinguishing the former from the later based on an over-dispersion criterion.

I do like the principle of the approach but the construction is only partly convincing. My main concerns are the following:

1 - I am not completely clear about the proof if identifiability. The proof of Thm 3.1. starts with "assume that the true causal ordering \pi^* is {1, 2, ...p}". To me, in a general oriented graphical model, there may be several true causal orderings. As the proof relies on induction, this should be clarified.

2 - The exploration of the DAG space is indeed a critical task and the authors simply suggest to use GLMlasso at each step to reduce the search space. The objective function of the GLMlasso is not necessarily consistent with the overdispersion criterion, so the global procedure is a bit heuristic.

3 - In the simulation study, I did not understand why all regression coefficients $\theta_{jk}$ are chosen to be negative (between -1 and -.7). This is allthemore surprising as the authors states that Yang requires this negativity constraint, while they do not. Does this choice make the problem easier?

4 - As for the performances, indeed they are good will large $n$, even for fairly large $p$, by I have been very surprised by the bad performances for $p = 10$ and $n = 500$ which seems to be a rather comfortable case. Do the authors have an idea why they fail in such a, apparently, easy situation?

Minor remarks:

1 - In the proof of Thm 3.1 should "Pa(j)/{1, 2, ..., m}" be "Pa(j)\{1, 2, ..., m}"?

2 - The threshold $c_0$ is likely to have some influence on the results, which is not explored in the simulation study.

Summary: The paper phrases an interesting problem in a clever and potentially efficient way. However the proposed algorithm is partly ad-hoc and the simulation is not fully convincing.

Author Feedback
Author rebuttal: We thank the reviewers for their efforts and comments for Paper 437. We are grateful that reviewers found the paper original, interesting, and noted that it tackles an important problem. There were a few questions raised by the reviewers which we clarify below in detail.

Rev #2 raises a question about notations in eq. 4.
We are thankful for Rev #2 for raising this issue. We agree that the notation of C_{jk} should be much clearer and we'll address this in the camera-ready version, if the paper is accepted. To clarify, the goal of our algorithm is to find the causal ordering and then the set of parents for each node. C_{jk} defines a set of candidate parents which is a superset of the actual parents. The subscript j runs from 1 to p and denotes the index of the element of the causal ordering we are searching for. k (which depends on j) denotes a node index in the set of candidate nodes for the j^th element of the causal ordering. Hence, C_{jk} is the set of candidate parents corresponding to the k^th node index for the j^th element of the causal ordering.

Rev #2 asks how the algorithm behaves under the different settings and model mismatch.
We have considered a lot of different settings such as different link functions, different ranges of parameters \theta and different in-degree. In short, our algorithm works well as long as Assumption 4.1 is satisfied. We will discuss and address model mismatch in the camera-ready version if the paper is accepted.

Rev #3 comments that our comparison methods in simulations are not appropriate for the Poisson count data.
We thank Reviewer 3 for his/her comments and feedback. We agree that more details are required to explain how the algorithms we present are adapted to count data. In short score-based methods are adapted by using the Poisson log-likelihood score. Constraint-based methods such as the PC algorithm are adapted using a conditional-independence test based on mutual information. We thank the reviewer for pointing this out and will include these details in the camera-ready version of the paper if it is accepted.

Rev #3 also comments that we do not link our DAG model to conditional independence.
We thank Reviewer 3 for pointing this out. We do not comment on it explicitly but it is implied by the factorization (see e.g. Lauritzen '96). We will clarify this point in the camera-ready version of the paper if it is accepted.

Rev #4 asks the clarity of the proof of identifiability condition.
We are grateful for the insightful questions raised by Rev #4. As Rev #4 mentions, the true causal ordering may not be unique. However, the mathematical induction is still valid for any of the valid causal orderings. For any true causal orderings, the first component satisfy the property that the expectation is same as variance otherwise overdispersed. Given the chosen first component of causal ordering the second elements of the true causal ordering hold the property that conditional expectation is equal to conditional variance and otherwise overdispersed. We will make this more explicit in the camera-ready version if the paper is accepted.

Rev #4 brings up a problem that an objective function of the GLMlasso is not necessarily consistent with the overdispersion criterion, so the global procedure is a bit heuristic.
We agree that the GLMLasso procedure is a bit heuristic since the GLMLasso procedure is not necessarily consistent. Due to space constraints, we do not address this issue. However in a longer version of the paper (to be posted on arxiv shortly), we prove that our heuristic is suitable when Assumption 4.1 is satisfied. We will add a comment to that effect in the camera-ready version if the paper is accepted.

Rev #4 raises a questions about our simulation settings.
On the first question regarding the choice of $\theta_{jk}$'s to be negative, we are ensuring Assumption 4.1 is satisfied. The choice of negative $\theta_{jk}$'s does make the problem easier since Assumption 4.1 is guaranteed to be satisfied however there are settings in which some $\theta_{jk}$'s are positive and Assumption 4.1 is satisfied.

On the second question regarding the $p=10$ and $d = 3$ case, our algorithm is working in some high dimensional settings. However, the algorithm needs a large sample size, especially when $d$ is large. As we mention, we need approximately $n = \Omega(\log p^{6d})$ which is extremely large when $d=3$. This problem does not only apply to our algorithm but also for other learning DAG algorithms such as PC, MMHC algorithms in which the edges are determined by a conditional independence test. We will make both of these points more explicit in the camera-ready version if the paper is accepted.

We would also like to thank Reviewers #5-7 for their positive reviews and feedback.